# A Simple Methodology to Develop Bifilar, Quadrifilar, and Octofilar Calculable Resistors

**Alepth H. Pacheco-Estrada** [1,*]ⓘ, **Felipe L. Hernandez-Marquez** [2]ⓘ, **Carlos D. Aviles** [2], **Carlos Duarte-Galvan** [3]ⓘ, **Juvenal Rodríguez-Reséndiz** [1]ⓘ **and Humberto Aguirre-Becerra** [4]ⓘ **and Luis M. Contreras-Medina** [4,*]ⓘ

[1] Faculty of Engineering, Autonomous University of Querétaro, Querétaro 76010, Mexico; juvenal@uaq.edu.mx

[2] Electromagnetic Measurements Direction, Centro Nacional de Metrología (CENAM), Querétaro 76246, Mexico; fhernand@cenam.mx (F.L.H.-M.); cdaviles61@gmail.com (C.D.A.)

[3] Faculty of Physical-Mathematical Sciences, Autonomous University of Sinaloa, Culiacán 80000, Mexico; carlos.duarte.galvan@uas.edu.mx

[4] Basic and Applied Bioengineering Group, Faculty of Engineering, Autonomous University of Querétaro, Querétaro 76265, Mexico; humberto.aguirreb@uaq.mx

[*] Correspondence: apacheco07@alumnos.uaq.mx (A.H.P.-E.); miguel.contreras@uaq.mx (L.M.C.-M.); Tel.: +52-442-2110500 (ext. 3428) (A.H.P.-E.)

**Abstract:** This paper describes the development of bifilar, quadrifilar, and octofilar Calculable Resistors (CRs). The research involves Evanohm-S and Isaohm wire heat treatment processes to achieve temperature coefficients less than $0.5\ \mu\Omega/\Omega/^{\circ}\mathrm{C}$ in the CR's wire resistance element, tests of different terminal–wire joining techniques, and construction aspects achieving a stability of less than $0.05\ \mu\Omega/\Omega/\mathrm{day}$. This kind of construction methodology has not been presented in detail in previous CR papers, and it is essential to accomplish the correct parameters of a CR. Without it, the development of a CR can take several months or even years. A comparison between CRs developed in this research and a CR from the Federal Institute of Metrology (METAS) in Switzerland was carried out. Measurement results between the 10 k$\Omega$ octofilar CR and the METAS 1 k$\Omega$ coaxial CR show an agreement better than $0.35\ \mu\Omega/\Omega$ through the audio-frequency range. Therefore, the octofilar CR can be used as an AC resistance reference with traceability to the quantum Hall resistance in DC.

**Keywords:** calculable resistor; frequency dependence; impedance metrology; reference standard; evanohm heat-treatment

## 1. Introduction

A CR is defined as a resistance standard with a calculable frequency dependence [1–3]. It has a known geometry that allows it to be modeled as a transmission line. Therefore, an equation describing the change of the resistance with the frequency can be determined. There are certain types of CR geometries, such as coaxial [1], bifilar [2], quadrifilar [2], and octofilar [3], whose main element consists of a thin NiCr alloy wire, configured as 4-terminal pair standards [4], and with appropriate heat treatment, it is possible to decrease its Temperature Coefficient (TC) below $1\ \mu\Omega/\Omega/^{\circ}\mathrm{C}$ [5].

The CR plays an essential role in the Farad traceability chain to the Quantum Hall Resistance (QHR) [6]. Through a DC resistance calibration using the QHR as a reference standard that only depends on fundamental constants, the CR is employed as a reference at frequencies of 1592 or 1542 Hz. The 100 k$\Omega$ resistors can then be used in a quadrature bridge to determine the values of the standard capacitors [6]. Since with the CRs, the frequency dependence of impedance standards with traceability to the QHR can be measured, they are used as references for high-accuracy electrical

measurement systems [7–9]. Applications with medical and technologic impact involve high-accuracy impedance measurements, such as electrical characterization of tissue [10,11], cell detection [12], supercapacitors developments [13], impedance matching in wireless power transfer systems [14], voltage-controlled inverters for electrical grids [15], and characterization of lead-acid batteries [16,17], only to cite a few examples. To ensure the certainty of the results of environmental research, like solar cell characterization [18], toxicity measurements of nanoparticles [19], or Nitrogen fertilizer monitoring in plants [20], the traceability to fundamental constants of their impedance measurements needs to be fulfilled. If the equipment used in the aforementioned applications do not realize accurate measurements, the implications could be to make wrong decisions, such as apply an excess of fertilizer to plants, which contaminate the soil and the disposable water to humans [21]. Likewise, the equipment that measures power factors implies traceability to impedance standards, therefore, having high accuracy measurement of power factors means to adopt better strategies to increase the efficiency of electrical energy use, which has a direct ambient impact [22].

The frequency performances of CR type coaxial, bifilar, quadrifilar, and octofilar resistors have been evaluated through international comparison. The results indicate that any of those geometries achieve the expected frequency behavior [23]. Nevertheless, their frequency dependence needs to be assessed to ensure no flaws during their construction [24].

Nowadays, at the Centro Nacional de Metrologia (CENAM) in Mexico, the capacitance measurements have traceability to the QHR reproduced at the Bureau International des Poids et Mesures (BIPM) in Paris, France. A set of CRs was developed to form part of the CENAM QHR—Farad traceability chain; one with a bifilar geometry of 1 kΩ, another with a quadrifilar geometry of 1 kΩ, and two with octofilar geometry of 10 kΩ. Figure 1 exhibits the CRs exploded views for the three types of geometries. The 10 kΩ CR will be used as a reference to calibrate 100 kΩ resistors at 1592 Hz. Then, they will be part of a quadrature bridge to determine the value of 1 nF capacitance standards [25]. Additionally, the set of CRs will be utilized as standards to improve the frequency impedance measurement capabilities.

During 2017, a 1 kΩ bifilar CR and a 10 kΩ octofilar CRs traveled to the Federal Institute of Metrology (METAS) in Switzerland. A set of 1:1 and 10:1 ratio measurements was performed using as a reference a METAS 1 kΩ coaxial CR. Some of the results are presented in [24] and are explained here in detail, together with the measurement uncertainty analysis and the complete description of the construction methodology of the CR.

Descriptions of the geometry and the model of the CRs have been described in previous works [1–3,25,26]. However, none of those describes details of the construction process. The geometry of the wire supports, the wire strain, and the terminal–wire joining method strongly affect the stability of a CR resistance value. The wire heat treatment of the CR is fundamental to reduce their TC below 1 $\mu\Omega/\Omega/°C$. Even if the CR is in a temperature-controlled environment within $\pm0.01$ °C, to get variations of less than 0.01 $\mu\Omega/\Omega$ in their reference value, its TC needs to be under 1 $\mu\Omega/\Omega/°C$.

High-resistive wires with diameters around 20 μm need to be employed to achieve nominal resistance values of 10 kΩ. The use of this type of thin wire makes each test time-consuming and challenging to perform. Therefore, obtaining the correct levels of TC and stability mentioned above without proper guidance, can take several months or even years. The purpose of this work is to serve as a guide and help to obtain the correct levels of temperature and stability of CRs in less time by using detailed procedures described in this paper and obtain CRs with an excellent performance that allows them to be used to measure the frequency dependence of impedance standards with traceability to QHR.

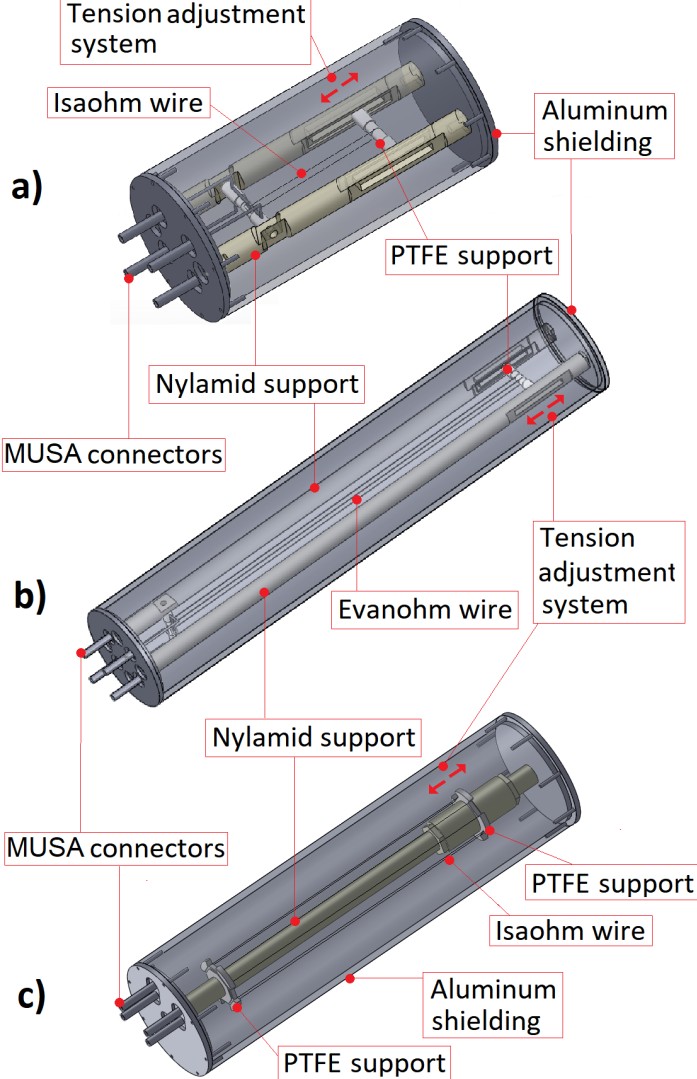

**Figure 1.** Assembly of CR pieces of: (**a**) 1 kΩ bifilar, (**b**) 1 kΩ quadrifilar, and (**c**) 10 kΩ octofilar.

Section 2 of this research describes the geometries and models of the CR type bifilar, quadrifilar and octofilar resistors developed. Likewise, a detailed procedure to perform heat treatment to NiCr alloy wires, methods to perform wires to terminal connections, and geometry considerations in the wire supports are presented. Section 3 exposes the results of the stability related to the terminal–wire joining techniques, the thermal treatment of two types of NiCr wire (Isaohm and Evanohm-S), and the stability related to the CR support geometry; at the end of the section, the measurement results between the CRs of CENAM and the METAS are presented. Finally, in Section 4, the discussion of this research is given.

## 2. Description of the Calculable Resistors

### 2.1. CRs Geometries and Model Descriptions

To model a CR as a transmission line, wire loops are placed in the middle of the cylindrical conductor shield in a way that each wire segment is located at the same distance from the shield axis. As shown in Figure 2a, the bifilar geometry consists of a single wire loop [27]. Figure 2b illustrates the quadrifilar geometry consisting of a double loop arranged so that the current flows in opposite directions in each wire segment [2]. As displayed in Figure 2c, the octofilar geometry is like

the quadrifilar geometry but with four-wire loops in series that tend to cancel magnetic fluxes [3]. The reason why there are geometries with more wire loops is that the total length of the resistor is reduced, allowing to develop standards with higher nominal resistance values. For example, if the length of a 1 kΩ bifilar CR is one meter (two meters of wire), the octofilar geometry with the same two-meter wire segment will be 25 cm.

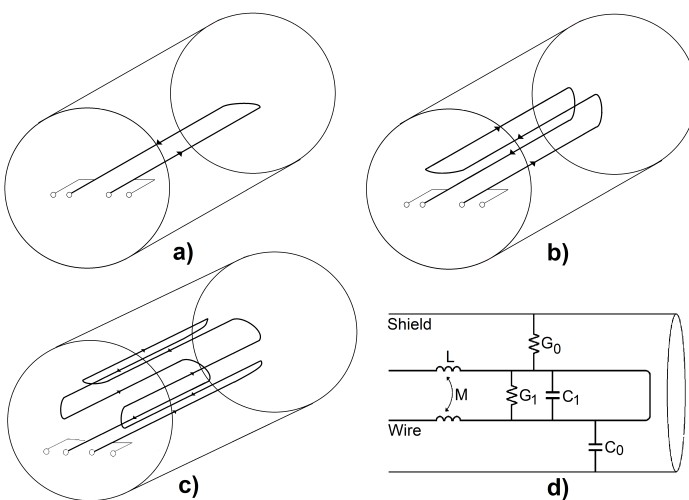

**Figure 2.** Types of calculable resistor geometries. (**a**) Bifilar geometry; (**b**) Quadrifilar geometry; (**c**) Octofilar geometry; (**d**) Parasitic impedance in a bifilar CR.

The frequency performance of a resistor depends on several factors, as has been correctly explained in previous works [2,3,25,28]; the presence of capacitance, inductance, and conductance between wire segments as well as among wire and shield, mostly affect the frequency dependence. As an example, Figure 2d exhibits the parasitic impedances present in a bifilar CR, where $L$ is the self-inductance of each wire segment, $C_0$ and $G_0$ are the total capacitance and conductance among the wire loop and the shield, $M$, $C_1$, and $G_1$ are the mutual inductance, capacitance, and conductance between wire segments respectively. For the quadrifilar and the octofilar geometries, the mutual inductance, capacitance, and conductance among each pair of wire segment needs to be taken into account; hence, the quadrifilar CR will have $M_1$ and $M_2$ mutual inductance, $C_1$ and $C_2$ capacitances, and $G_1$ and $G_2$ conductances between wires, and the octofilar CR will have $M_1$, $M_2$, $M_3$, and $M_4$ mutual inductances, $C_1$, $C_2$, $C_3$, and $C_4$ capacitances, and $G_1$, $G_2$, $G_3$, and $G_4$ conductances between wires

According to the calculations described in [2,3], the models that describe the change in resistance as a function of the angular frequency ($\omega$) due to the parasitic impedances in the CRs are:

$$\frac{R_{ac}}{R_{dc}} = \left\{ 1 + \frac{R}{6}(G_0 - 2G_1) + \left(\frac{\omega}{R}\right)^2 (L - 2M_1)^2 + \frac{\omega^2 R^2}{720}\left[15C_0^2 - (C_0 + 4C_1)^2\right] \right\} \tag{1}$$

for the bifilar geometry presented in Figure 2a,

$$\frac{R_{ac}}{R_{dc}} = \left\{ 1 + \frac{R}{6}(G_0 - 5G_1 - 3G_2) + \left(\frac{\omega}{R}\right)^2 (L - 8M_1 + 4M_2)^2 \right. \\ \left. + \frac{\omega^2 R^2}{11520}\left[240C_0^2 - (C_0 + 16C_1)^2 - 15(C_0 + 8C_1 + 8C_2)^2\right] \right\} \tag{2}$$

for the quadrifilar geometry showed in Figure 2b, and

$$\frac{R_{ac}}{R_{dc}} = \left\{ 1 + \frac{R}{192}(17G_0 - 128G_1 - 144G_2 - 128G_3 - 48G_4) + \right.$$

$$\left(\frac{\omega}{R}\right)^2 (L - 16M_1 + 16M_2 - 16M_3 + 8M_4)^2 + \frac{\omega^2 R^2}{737280}\left[1920C_0^2 - 4(C_0 + 32C_1 + 32C_3)^2 \right. \quad (3)$$

$$\left. \left. - 60(C_0 + 16C_1 + 32C_2 + 16C_3)^2 - 15(C_0 + 16C_1 + 16C_2 + 16C_3 + 16C_4)^2\right]\right\}$$

for the octofilar geometry exhibit in Figure 2c.

The value of the parasitic impedances present in the CRs is calculated through their dimensions. For self-inductances and mutual inductance values, Rosa's and Grover's formulas are utilized [29]. The method described by Zhang [30] is employed to define capacitance values. As described by Bohacek and Wood [3], the conductances can be calculated from capacitances if the permittivity and the conductivity of the medium are known. Tables 1 and 2 summarize, respectively, the dimensions of the three types of CRs developed in this research and the calculated values of their parasitic impedances. For the three types of CRs, the shields are made of aluminum tubes, and the dielectric is air. Trying to achieve the lowest resistance changes due to frequency, the models of Equations (1)–(3) were typed in a Matlab program to evaluate the CR's frequency behavior as a function of the distances between adjacent wires, shield diameter, shield thickness, and wire diameter. In this way, the geometric parameters of the RCs were selected.

There are other factors such as Eddy currents that influence the frequency dependence of a CR [2], and the skin effect that changes the resistance of a cylindrical conductor due to the tendency of the current to flow through the wire surface as frequency increases.

The increase in resistance owing to the skin effect is calculated from Equation (4) [3,28]. As was established in [2], there are no significant Eddy current losses in each wire due to the proximity of the others. The increase in resistance due to the Eddy currents induced in the cylindrical shield is expressed by Equation (5) [2].

$$\frac{R_{ac}}{R_{dc}} = 1 + \frac{\omega^2}{192}\left(\frac{\mu_0\mu_r}{\pi r}\right)^2 \tag{4}$$

$$\frac{R_{ac}}{R_{dc}} = 1 + \frac{2\mu_0 m\omega}{\pi r}\sum_{n}^{\infty}\frac{1}{m^2 + n^2}\left(\frac{b}{a}\right)^2 \tag{5}$$

where $m = \frac{\omega\mu_0\mu_s as}{2\rho_s}$, $\mu_0$ is the permeability of vacuum, $\mu_r$ is the relative permeability of the wire, $r$ is the resistance per unit of length of the wire, $\omega$ is the angular frequency in $rad/s$, $b$ is the distance between wires, $a$ is the internal shield radio, $s$ is the shield thickness, $\mu_s$ is the relative permeability of the shield, and $\rho_s$ is the resistivity of the shield.

**Table 1.** Dimensions of bifilar, quadrifilar, and octofilar CRs.

| Parameter | Symbol | Bifilar | Quadrifilar | Octofilar |
|---|---|---|---|---|
| Shield thickness | s | $6.0 \pm 0.1$ mm | $6.0 \pm 0.1$ mm | $6.0 \pm 0.1$ mm |
| Shield diameter | a | $102.1 \pm 0.1$ mm | $102.1 \pm 0.1$ mm | $102.1 \pm 0.1$ mm |
| Wire diameter | r | $21.66 \pm 0.07$ mm | $57.34 \pm 0.83$ mm | $21.66 \pm 0.07$ mm |
| Folded length of wire | l | $119 \pm 0.5$ mm | $443 \pm 1$ mm | $291 \pm 1$ mm |
| Distance between adjacent wires | d | $9.4 \pm 0.5$ mm | $6.7 \pm 0.5$ mm | $16.1 \pm 0.5$ mm |
| Distance between shield axis and wires axis | b | $4.7 \pm 0.5$ mm | $4.7 \pm 0.5$ mm | $20.8 \pm 0.5$ mm |

**Table 2.** Parasitic impedances values of bifilar, quadrifilar and octofilar CRs.

| Impedance | Bifilar | Quadrifilar | Octofilar |
|:---:|:---:|:---:|:---:|
| $M_1$ | 0.057 µH | 0.345 µH | 0.117 µH |
| $M_2$ | - - - - - | 0.315 µH | 0.086 µH |
| $M_3$ | - - - - - | - - - - - | 0.074 µH |
| $M_4$ | - - - - - | - - - - - | 0.070 µH |
| $L$ | 0.880 µH | 4.156 µH | 2.360 µH |
| $C_0$ | 1.304 pF | 4.410 pF | 10.284 pF |
| $C_1$ | 0.164 pF | 0.862 pF | 0.278 pF |
| $C_2$ | - - - - - | 0.537 pF | 0.084 pF |
| $C_3$ | - - - - - | - - - - - | 0.049 pF |
| $C_4$ | - - - - - | - - - - - | 0.042 pF |
| $G_0$ | 9.981 aS | 52.823 aS | 78.692 aS |
| $G_1$ | 1.257 aS | 6.600 aS | 1.666 aS |
| $G_2$ | - - - - - | 4.110 aS | 0.640 aS |
| $G_3$ | - - - - - | - - - - - | 0.378 aS |
| $G_4$ | - - - - - | - - - - - | 0.320 aS |

## 2.2. Evanohm-S and Isaohm Wires to Terminals Connection

To develop CRs with nominal values of 10 kΩ and 1 kΩ with low-temperature coefficients, the wire must be of a high-resistive alloy, able to be heat-treated to decrease its TC near zero. Therefore, the wires chosen for this purpose were Evanohm-S wire (72% Ni, 20% Cr, 4% Mn, 3% Al and 1% Si) of 57.3 µm of diameter and a resistivity of 145.6 µΩ·cm, and Isaohm wire (74.5% Ni, 20% Cr, 0.5% Mn, 3.5% Al, 0.5% Fe and 1% Si) of 21.7 µm of diameter and a resistivity of 154.8 µΩ·cm.

In the development of a CR, the joining between the thin wire and the terminals is the first stage to be considered. To achieve a joining that does not cause undesirable instability and dispersion, three methods for joining the Evanohm-S wire with a 14-gauge copper terminal were tested.

The first method was to use a 14-gauge copper tube that crimps the wire on the inside, using the edges of crimp clamps with a force of ≈2 kN. In the second method, a tin/lead-free solder was used employing a standard soldering station, using a temperature of ≈380 °C. The last method was to use a homemade spot welder with 90 W of power and parallel copper electrodes with a flat tip made of 14-gauge copper wire. The three methods were tested using Evanohm-S wire segments of 16 cm of length and copper terminals, placed inside an aluminum box on an oil bath with temperature control of 23.00 °C ± 0.01 °C, measuring them as a four-terminal resistance with an Agilent 3458A multimeter.

## 2.3. Wire Heat Treatment Process

The next step to be considered in the development of CRs is the heat treatment of the wire to decrease their TC. In this research, Evanohm-S [31] and Isaohm [32] wire spools with previous heat treatment by the manufacturer were tested; the properties of these kinds of alloys make it possible to realize a heat treatment to modify their TC. As was reported by Starr [5], regardless of the heat-treating temperature used in the wire, its TC decreases from the positive value of 50 µΩ/Ω/°C, passes through zero, and eventually reaches a minimum value of around −35 µm of diameter and a resistivity of 154.8 µΩ/Ω/°C; if heat treatment continues, the TC starts becoming less negative, passing through zero again, and continue increasing slowly. At higher temperatures, less time is needed to achieve a change in the TC; however, these parameters depend on the NiCr alloy wire and on their gauge. This behavior is represented by Figure 3. Hence, to decrease the TC of NiCr alloy wires, there are two

independent variables, temperature and time of the heat treatment. An explanation of the relationships between the heat treatment time and the temperature is described by Starr in [5].

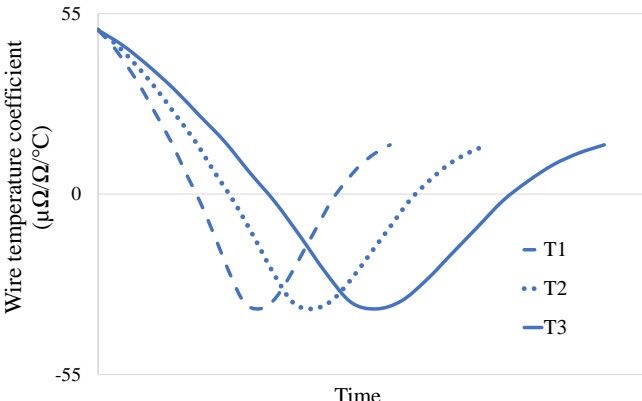

**Figure 3.** Graph of the change in Evanohm wire TC at temperatures T1 > T2 > T3. The time scale depends on the type of NiCr alloy wire and their gauge.

The process to achieve a TC below 1 $\mu\Omega/\Omega/°C$ consists of the following steps: (A) Measuring the TC of a wire segment of the spool without heat treatment. (B) Determine where the TC is plotted in Figure 3 graph (C) Perform heat treatment to a non-heat-treated wire segment. (D) Take a section of the heat-treated wire and measure their TC. (E) Depending on the change of the TC, analyze if more or less treatment time or temperature is needed to achieve the required TC. (E) Repeat steps B to D until the desired TC is obtained. A suitable time and temperature to carry out the first heat treatment might be 1 hour at 400 °C.

To measure the wire TC, copper terminals were placed at the end of the tested wire segments using the spot-welding technique; placing the wire segment inside an aluminum shielding submerged in an oil bath with programmable temperature control, a set of 1 °C steps, every four hours, from 20 °C to 26 °C, were performed. In parallel, the resistance of the wire was measured as a four-terminal terminal resistor through an Agilent 3458A meter.

The heat treatment was carried out by winding three meters of wire on a glass support. Then the support was placed in a quartz case inside a homemade temperature oven able to reach temperatures up to 450 °C.

*2.4. Wire Strain*

Although the CRs are used as a transfer standard between DC and AC measurements, the stability needs to be enough to handle the DC value as a reference during the AC measurement. This stability is achieved when there is a low wire strain present on the resistance element. However, if the wire is too loose, the CR geometry is not well defined. In this research, considerations in the wire support geometries are presented to obtain stability better than 0.05 $\mu\Omega/\Omega/$day.

It is also necessary to avoid any kinks in the CR wire segment. The kinks need a high-strain to correct and accomplish the CR's geometry, causing drifts higher than 10 $\mu\Omega/\Omega/$h, which makes it impossible to measure the CR's value with a high-accuracy measurement system, like a cryogenic current comparator. Other difficulties are wire supports with right angles. If a moderate strain is applied to the wire, the natural geometry of the wire segments does not allow the sections to be parallel, and if it has a high-strain, the drift goes up to $\mu\Omega/\Omega/$hour. The solution is the use of rounded and smooth supports. Figure 4 presents the supports used for the bifilar, and octofilar geometry with their strain adjustment systems. The quadrifilar CR has the same strain adjustment system as the bifilar CR. The only difference is that the quadrifilar PTFE support has two guide grooves instead of only one. For the bifilar and the quadrifilar geometries, the position of the top support is adjusted by unscrewing

the nylamid screws that hold the PTFE support to the nylamid supports. The octofilar CR has threaded supports that hold the PTFE wire supports, allowing to move it vertically to adjust the strain.

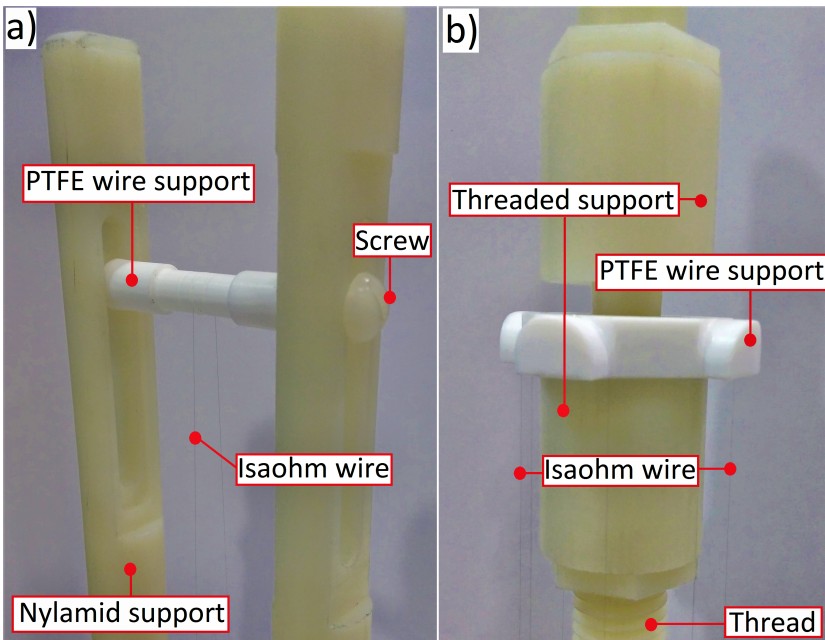

**Figure 4.** CR's wire supports with strain adjustment system, for: (**a**) the bifilar CR, and (**b**) the octofilars CRs.

To achieve stability below 0.05 μΩ/Ω/day, the strain in the wire needs to be as low as possible. The procedure consists of manually adjusting the strain, until all the wire segments are parallel within ±0.5 mm, measuring it with a calibrated Vernier, ensuring its perpendicularity with the wire segments during the measurements. Then, the stability of the resistance value is measured, and if it is higher than 0.1 μΩ/Ω/day, the strain is readjusted again.

A commercial 6000B Measurements International bridge was used to measure the stability of the CRs. The measurement system gives the 1:1 or 10:1 ratio between the measured CR and a 10 kΩ ESI SR104 standard resistor. Because of the reference standard resistor has stability better than 0.1 μΩ/Ω/year, the stability of the measured CR was not affected by it.

*2.5. Bifilar and Octofilar Frequency Evaluation*

During 2017, the bifilar and one of the octofilar CRs were sent to METAS, which is the National Metrology Institute (NMI) of Switzerland [24]. To not damage the integrity of the wires and their connections to their terminals, before sending the CRs to Switzerland, their wire strains were wholly removed moving the PTFE support, one centimeter down. At METAS, the strain in the wires was re-adjusted through the strain adjustment systems displayed in Figure 4, and after one month, the two standards of CENAM had stabilities better than 0.2 μΩ/Ω/day. The CRs of CENAM are contained in individual temperature-controlled enclosures to avoid any temperature effect in the resistance value. The enclosures ensure temperature stability of ±0.015 °C, which implies variations of the resistance value due to temperature changes of less than 0.02 μΩ/Ω.

Because the METAS 1 kΩ coaxial CR frequency dependence has been validated through different comparisons [23,33], it was chosen as the reference standard for the evaluation of CENAM CRs. Table 3 listed the three CRs involved in the performed measurements.

**Table 3.** Standards involved in the frequency evaluation of the CRs

| Name | Type | Nominal Value | NMI | Wire/Dielectric |
|---|---|---|---|---|
| Haddad1000 1201 | Coaxial | 1 kΩ | METAS | Evanohm/air |
| RC-1k-D | Bifilar | 1 kΩ | CENAM | Isaohm/air |
| RC-10k-D5e | Octofilar | 10 kΩ | CENAM | Isaohm/air |

To perform the measurements between CENAM and METAS CRs, a Coaxial Bridge developed at METAS was utilized [33]. This bridge is able to perform automatic measurements of four-terminal-pair impedance standards with 1:1 and 10:1 ratios. To provide the voltage ratio, the bridge uses a double-screened ratio transformer (RT). Using a reversal measurement technique in the 1:1 ratio measurements, the effect of the RT error, and the effect of the cables used to connect the standards to the bridge are eliminated. For the 10:1 ratio, the frequency dependence of the RT error was measured using, as a reference, coaxial CRs with nominal values of 12,906 Ω and 1290.6 Ω. Because the time required for a frequency sweep from 50 to 20 kHz does not take more than 2.5 hours, including the reversal measurements for the 1:1 ratio, the short term stability of the standards have an impact on the results of less than 0.025 μΩ/Ω.

## 3. Measurements Results

### 3.1. Frequency Performance of the Bifilar, Quadrifilar and Octofilar CRs

Figure 5 display the calculated frequency performance of the bifilar, quadrifilar, and octofilar CRs using the dimensions from Table 1 in Equations (1)–(3), respectively, and taking into account the change in resistance due to the eddy current effects of Equations (4) and (5). The change in the resistance concerning its nominal value increases with the number of wire segments. However, the change in resistance at frequencies under 5 kHz is less than 0.03 μΩ/Ω for any of the three types of CRs. Hence, at 1592 Hz, the resistance value could be considered equal to the DC calibrated value within 0.01 μΩ/Ω.

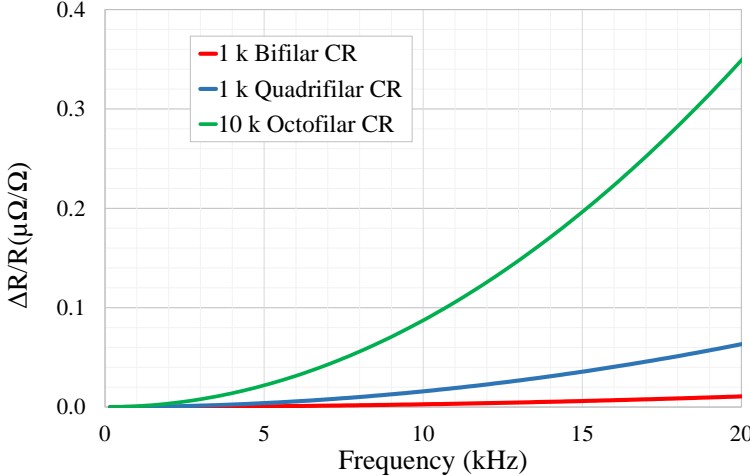

**Figure 5.** Calculated frequency performance of the bifilar, quadrifilar, and octofilar CRs.

### 3.2. Methods for Joining Evanohm-S Wire To Terminals

The results for the three methods for joining Evanohm-S to a terminal are shown in Figure 6. It is evident that the tin soldering method does not achieve a proper joining, causing changes in the measured resistance values up to 90 μΩ/Ω in less than one day. The method of introducing the wire in a copper tube and crimping it manifested a 0.25 μΩ/Ω/hour drift. The drift could be due

to the mechanical stress of the tight section of the wire that changes the shape of the wire in the junction. For this reason, the methods that involve mechanical stress in the wire should be avoided. Finally, joining with a spot welder does not present drift or changes in the measured resistance value. The standard deviation of the measurements with the last method is less than 0.5 $\mu\Omega/\Omega$ and is due to the measurement system in the 100 $\Omega$ range.

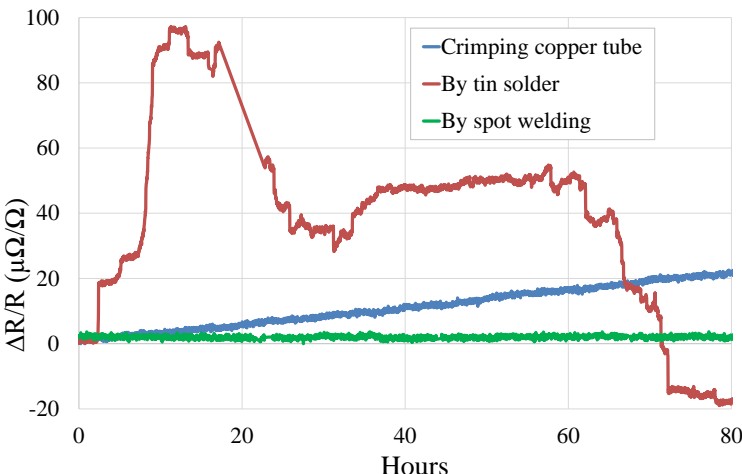

**Figure 6.** Graph of the stability in the measured resistance value of a 16 cm Evanohm-S wire with terminals joined through tin solder, crimped copper tube, and spot welder.

Through the results, the method of joining the wire by a spot welder was chosen in this work for the development of CRs. This method also has the advantage of using it with an Evanohm terminal to avoid EMFs in the terminal–wire junction.

### 3.3. Evanohm-S and Isaohm Heat Treatment

The process for the Evanohm-S and Isaohm wires to reach a TC under 1 $\mu\Omega/\Omega/°C$ are described in Figure 7. Table 4 shows the steps to achieve the final heat treatment for the two types of wires. In all the attempts, the time to reach the maximum heat treatment temperature was 6 h, and the time to reach room temperature (23 °C) was 19 h. For all the performed heat treatments, the TCs were located on the first slope of the graph of Figure 3. According to [5], this allows us to have two paths to reach a TC near to zero: using the negative slope or using the positive slope after the −30 $\mu\Omega/\Omega/°C$ TC.

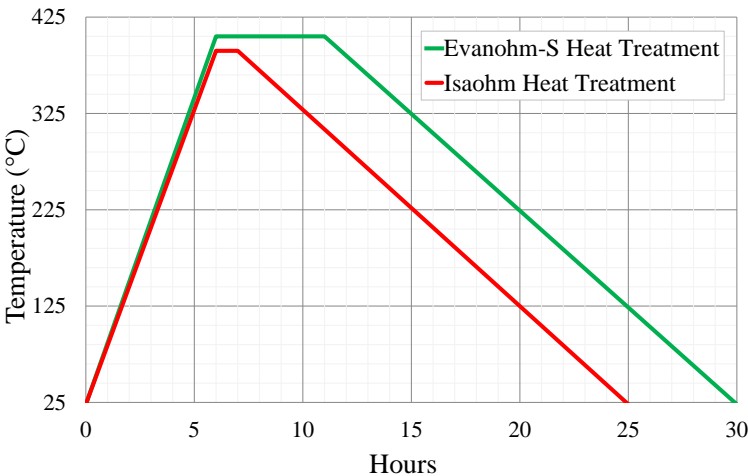

**Figure 7.** Heat treatment process to achieve TCs under 1 $\mu\Omega/\Omega/°C$.

**Table 4.** Steps to achieve the correct heat treatment.

| Isaohm Wire | | Evanohm-S Wire | |
|---|---|---|---|
| **Heat Treatment** | **Measured TC (µΩ/Ω/°C)** | **Heat Treatment** | **Measured TC (µΩ/Ω/°C)** |
| No heat treatment | 4.9 | No heat treatment | 6.8 |
| 5 h at 405 °C | −11.1 | 15 h at 405 °C | −3.6 |
| 1 h at 405 °C | −3.2 | 12 h at 405 °C | −2.2 |
| 1 h at 395 °C | −0.9 | 5 h at 405 °C | 0.3 |
| 1 h at 390 °C | −0.1 | | |

### 3.4. Stability of the CR's Resistance Value

As observed in Figure 8, the stability of the three types of CR with a wire strain that accomplishes the geometry of the CRs, are less than 0.03 µΩ/Ω/day. The measurements were made using a MIL 6000B bridge, with an applied power of 10 mW and within a temperature-controlled environment of ±0.02 °C. The measurement system gives the 1:1 or 10:1 ratio between the measured CR and a 10 kΩ ESI standard resistor. Because the reference standard resistor has stability better than 0.1 µΩ/Ω/year, the stability of the measured CR was not affected.

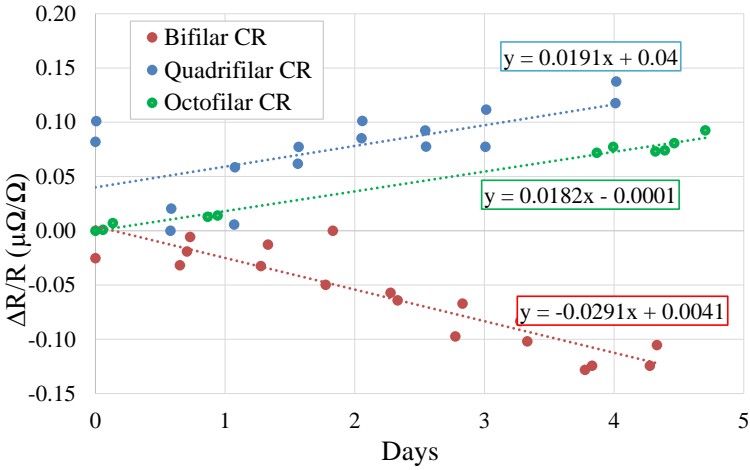

**Figure 8.** Short term stability of the bifilar, quadrifilar, and octofilar CRs.

### 3.5. Bifilar and Octofilar Frequency Evaluation

The frequency dependence evaluation of the CRs was done by calculating the relative difference between the measured and the calculated equivalent parallel resistance ratio of the CRs. Figure 9 shows the results for the 10:1 ratio between the RC-10k-D5e and the HADDAD1000-1201, from 300 Hz to 20 kHz. The ratio measurement results are in agreement with the computed ratio within the expanded uncertainty (k = 2.0). As seen in Figure 10, for the entire measured frequency range, the principal uncertainty components for the 10:1 ratio measurements are related to the RT error, the cable effects, the main balance resolution and the auxiliary balances of the bridge; for frequencies higher than 5 kHz, the main balance injection has a significant uncertainty contribution. The complete description of the uncertainty sources can be found in [33].

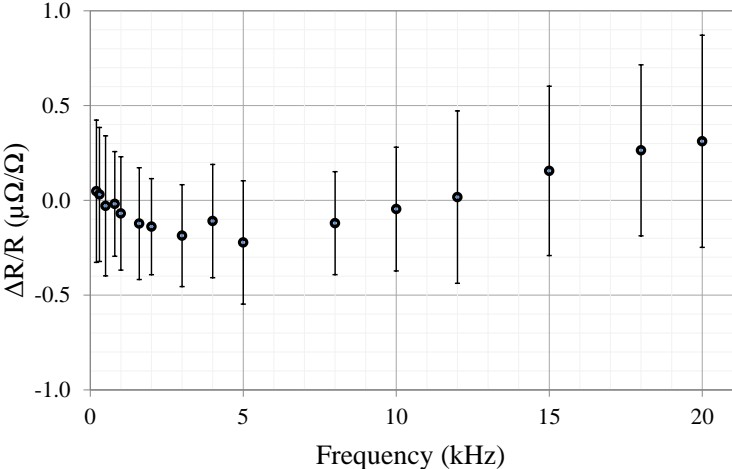

**Figure 9.** Graph of the relative difference between the measured and the calculated frequency dependence of the equivalent parallel resistance of the RC 10K-D5e and the HADDAD1000-1201 10:1 ratio, as a function of frequency [24].

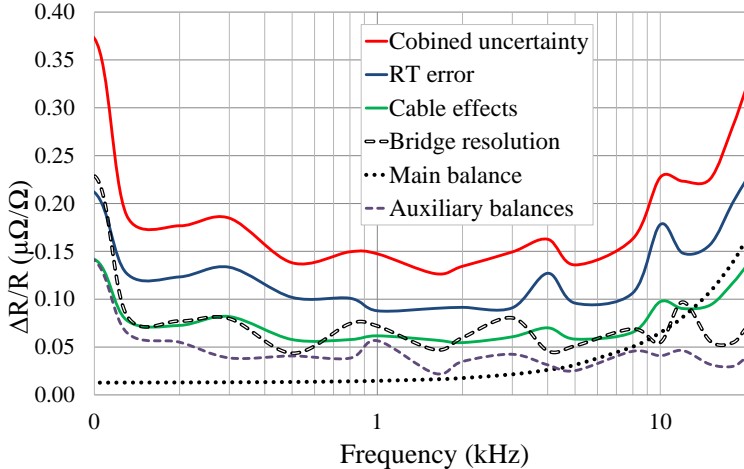

**Figure 10.** Standard uncertainty components (k = 1) of the 10:1 ratio measurement between the RC-10K-D5e and the HADDAD1000-1201.

Figure 11 presents the results for the 1:1 ratio between the RC-1k-D and RC-10k-D5e, and the 10:1 ratio between RC-1k-D and HADDAD1000-1201. In both measurements, a linear frequency dependence of about −0.16 μΩ/Ω/kHz was observed. This linear frequency performance is not expected in the model of CRs of any geometry. Nevertheless, this similar linear frequency behavior was found on CRs and was reported in [26,34].

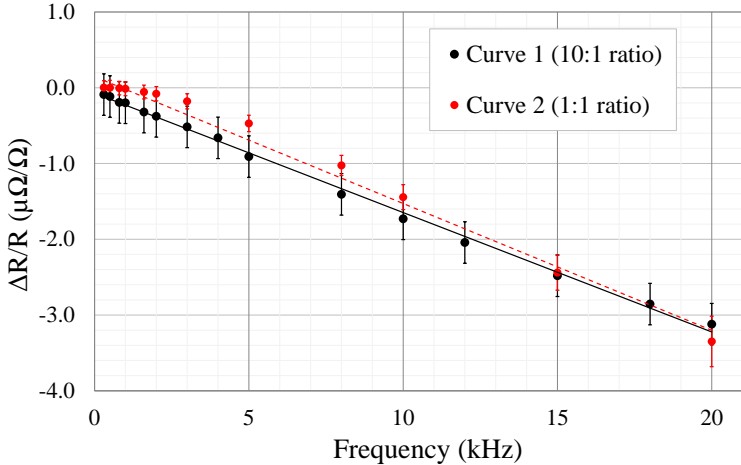

**Figure 11.** Graph of the relative difference between the measured and the calculated frequency dependence of the equivalent parallel resistance of CR-1k-D, as a function of frequency. Curve 1 using the CR-10k-D5e as reference (ratio 10:1, black points), Curve 2 using the Haddad1000 1201 as reference (ratio 1:1, red points) [24].

Some possible causes for this behavior are the bridge with which the measurements were made, the CRs dielectric, and the terminal–wire joining. Nevertheless, the measurement system is discarded because the bridge has been previously characterized by the same CRs employed in the international EURAMET comparison and no linear frequency dependence was found [23]. The dielectric in the CRs is not the reason because all the CRs have air as dielectric and were exposed to the same environmental conditions, and the linear frequency dependence is observed only when RC-1k-D is used. The most likely reason is located in the wire or the terminal–wire connections [26], due to damage caused by a very strong blow to the CR container during transport.

## 4. Discussion

The results of this research establish a methodology to develop CRs of bifilar, quadrifilar, and octofilar geometries with appropriate TC and stability. Therefore, low type A uncertainties are achieved in DC calibration with high-accuracy measurement systems, which allows the CR to be employed as references at different frequencies.

Three methods join Evanohm-S wire and copper terminals were evaluated. The results of the spot welding method offer measured wire resistance values without instability or drift over 0.05 $\mu\Omega/\Omega$/day. The same results were obtained using this method with Evanohm terminals and with Isaohm wire. Furthermore, the use of round supports ensures a CR's geometry with an appropriate mechanical strain on the wire. Therefore, the models of Equations (1)–(3) and  are utilized to determine the change in the CR's resistance value as a function of the frequency.

Previous CR research makes a profound description of the model for the change in resistance in functions of the frequency. However, none of them describe the construction methodology to achieve the appropriate TCs and stabilities to use the CR with uncertainties contributions of less than 0.05 $\mu\Omega/\Omega$ detailed in this research.

A methodology to perform heat treatment to Evanohm-S and Isaohm wires were presented. The results indicate that it is possible to use it in any NiCr alloy wire to decrease its TC to less than 0.5 $\mu\Omega/\Omega/°C$. However, it is essential that before starting the heat treatment tests, the method to join the NiCr wire to a terminal allows measuring the temperature coefficient without instabilities.

As seen in Figure 5, with the dimensions of Table 1, the change in the resistance value of the CRs at frequencies under 2 kHz could be considered as equal to the DC value with an uncertainty of less than 0.01 $\mu\Omega/\Omega$. Nevertheless, it is better to use the relative difference between the measured and

calculated frequency dependence between different CRs as the uncertainty due to the change in the resistance value.

The frequency dependence of CENAM's 10 kΩ octofilar CR and METAS's 1 kΩ coaxial CR are in good agreement with the calculated frequency dependence. These results allow using the RC 10K-D5e as a reference in the traceability chain between the QHR and the capacitance measurements at CENAM. However, the results obtained with the RC-1k-D bifilar CR show unexpected linear behavior. The cause for this behavior can be due to damage in the wire to terminal connections during transport from Mexico to Switzerland. Therefore, the wire in RC-1k-D will be replaced to evaluate if the linear frequency dependence disappears. Nevertheless, the spot-welding process, as the cause of the linear frequency performance, cannot be discarded. For this reason, before replacing the RC-1k-D wire, the parameters of the process, like the power application time and the mechanical strain between electrodes need to be controlled.

The authors are working on the development of a coaxial impedance bridge, similar to the one used at METAS for the frequency evaluation of the CRs. This measurement system will allow us to rule out the spot welding process as a cause for the linear frequency behavior.

**Author Contributions:** Conceptualization, F.L.H.-M., L.M.C.-M. and A.H.P.-E.; methodology, F.L.H.-M. and A.H.P.-E.; validation, F.L.H.-M., A.H.P.-E., C.D.A., L.M.C.-M., C.D.-G., H.A.-B. and J.R.-R.; formal analysis, A.H.P.-E., F.L.H.-M. and L.M.C.-M. investigation, F.L.H.-M. and A.H.P.-E.; resources, F.L.H.-M. and L.M.C.-M.; data curation, A.H.P.-E.; writing–original draft preparation, A.H.P.-E., J.R.-R. and L.M.C.-M.; writing–review and editing, F.L.H.-M., A.H.P.-E., C.D.A., L.M.C.-M., C.D.-G., H.A.-B. and J.R.-R.; visualization, F.L.H.-M., A.H.P.-E., C.D.A., L.M.C.-M., C.D.-G., H.A.-B. and J.R.-R.; supervision, F.L.H.-M., L.M.C.-M. and A.H.P.-E.; project administration, F.L.H.-M. and L.M.C.-M. ; funding acquisition, F.L.H.-M. All authors have read and agreed to the published version of the manuscript.

**Funding:** The collaboration stay at METAS Switzerland was funded by Physikalisch-Technische Bundesanstalt (PTB). The founding sponsors had no role in the design of the study.

**Acknowledgments:** The authors would like to thank Fréderic Overney from METAS for their support and guidance during the collaboration stay, and to Randolph E. Elmquist from the National Institute of Standard and Technology (NIST) for his guidance in the development of the CRs. Furthermore, the authors would like to thanks the Centro Nacional de Metrologia (CENAM) to provide all the equipment, material, and facilities used in this research.

**Conflicts of Interest:** The authors declare no conflict of interest.

## Abbreviations

The following abbreviations are used in this manuscript:

CR    Calculable Resistor
TC    Temperature Coefficient

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
