# Peer review of "A Simple Methodology to Develop Bifilar, Quadrifilar, and Octofilar Calculable Resistors"

_applsci, doi:10.3390/app10051595_

Round 1
Reviewer 1 Report
The article written by Pacheco and coautors reports on a simple method to fabricate calculable resistors. The article is well written, the fabrication and the results are well presented and discussed. The clear description of the fabrication steps adds great value to the work. I suggest the publication of this manuscript in Applied Sciences.
I have only a few comments/questions:
- table 1: why the diameter of the wire and the fistance between adjacent wires used for the quadrifilar CR differ from the other two configurations?
- Line 177, page 7: how do you measure the angle of the wires in order to achieve an accuracy in the of +-0.5 mm?
Author Response
Dear Reviewer,
We are pleased to resend you the entitled article "A Simple Methodology to Develop Bifilar, Quadrifilar, and Octofilar Calculable Resistors", in the Special Issue "Experimental Mechanics, Instrumentation and Metrology" of the Applied Sciences journal.
The authors appreciate the corrections and hope that the answers enlisted below fulfill the expectations. We considered every suggestion as a valuable opportunity to improve and enrich our work enormously. Your observations are highlighted using a bold black font, our replies using a blue font and changes in existing sentences have been highlighted in color.
- table 1: why the diameter of the wire and the distance between adjacent wires used for the quadrifilar CR differ from the other two configurations?.
A: Thank you for your comment. At the end of the first paragraph of page 5 (from line 113 to 116), the next extra lines were incorporated to clarify why the distances between adjacent wires and the wire diameters were chosen:
“Trying to achieve the lowest resistance changes due to frequency, the models of equations 1, 2, and 3 were typed in a Matlab program to evaluate the CR’s frequency behavior in function of the distances between adjacent wires, shield diameter, shield thickness, and wire diameter. In this way, the geometric parameters of the RCs were selected.”
- Line 177, page 7: how do you measure the angle of the wires in order to achieve an accuracy in the of +-0.5 mm?.
A: Thank you for your observation. To explain how to achieve the mentioned accuracy, the paragraph of line 190 to 194 was modified as follows:
“The procedure consists of manually adjusting the strain, until all the wire segments are parallel within ± 0.5 mm, measuring it with a calibrated Vernier, ensuring its perpendicularity with the wire segments during the measurements. Then, the stability . . . .”

Reviewer 2 Report
Applied Sciences
Manuscript Number: applsci-702430
Title: A Simple Methodology to Develop Bifilar, Quadrifilar, and Octofilar Calculable Resistors
Authors: Alepth H. Pacheco-Estrada, Luis M. Contreras-Medina, Felipe L. Hernandez-Marquez, Carlos D. Aviles, Humberto Aguirre-Becerra, Carlos Duarte-Galvan, Juvenal Rodríguez-Reséndiz
Paper „ A Simple Methodology to Develop Bifilar, Quadrifilar, and Octofilar Calculable Resistors " contains a lot of information about the development of Calculable Resistors (CR). In my opinion the paper could be published but needs corrections and complements which will raise quality of the paper.
The authors should address the following points:
1/ Introduction and Conclusion should be rewritten. The potential environmental impact should be included in the Introduction.
2/ The scale in figure 3 (time) should be added.
3/ The quality of Figure 4 should be improved. Photography could be more readable.
4/ In the conclusion, the spot welding process as a reason for the linear frequency behavior should be more emphasized.
Author Response
Dear Reviewer,
We are pleased to resend you the entitled article "A Simple Methodology to Develop Bifilar, Quadrifilar, and Octofilar Calculable Resistors", in the Special Issue "Experimental Mechanics, Instrumentation and Metrology" of the Applied Sciences journal.
The authors appreciate the corrections and hope that the answers enlisted below fulfill the expectations. We considered every suggestion as a valuable opportunity to improve and enrich our work enormously. Your observations are highlighted using a bold black font, our replies using a blue font and changes in existing sentences have been highlighted in color.
our work
- Introduction and Conclusion should be rewritten. The potential environmental impact should be included in the Introduction.
A: Thank you for your comment. We agree with your statement. A Calculable Resistor is a standard that gives traceability to impedance measurements at different frequencies to the Quantum Hall Resistance (fundamental constants). The impacts in all the fields of this tool are in the applications that involve impedance measurement systems or impedance standards.
To highlight the environmental impact, a few lines (from line 31 to 39) were included in the introduction as follow:
“To ensure the certainty of environmental researches, like solar cell characterization [18], toxicity measurements of nanoparticles [19], or Nitrogen fertilizer monitoring in plants [20], the traceability to fundamental constants, of their impedance measurements needs to be fulfilled. If the equipment used in the aforementioned applications, do not realize accurate measurements, the implications could be to make wrong decisions, such as apply an excess of fertilizer to plants, which contaminate the soil and the disposable water to humans [21]. Likewise, the equipment that measures power factor implies traceability to impedance standards, therefore, having high accuracy measurement of power factor, mean to adopt better strategies to increase the efficiency of electrical energy use, which has a direct ambient impact [22].”
- The scale in figure 3 (time) should be added.
A: Thank you for your comment. The time scale is not presented because it is different for each wire. The intention to include the graph of figure 3 is to describe the properties of the alloy. With this information, together with the wire heat treatment process, the temperature coefficient can be reduced to less than 1 parts per million / °C.
To clarify why the time scale is not presented in figure 3, the description of the figure was corrected as follow:
“Figure 3. Graph of the change in Evanohm wire TC at temperatures T1>T2>T3. The time scale depends on the type of NiCr alloy wire and their gauge.”
Also, in line 153, the next phrase was incorporate:
. . . .At higher temperatures, less time is needed to achieve a change in the TC; however, these parameters depend on the NiCr alloy wire and on their gauge. . . .
- The quality of Figure 4 should be improved. Photography could be more readable.
A: Thank you for your comment. Another photograph from inside the CRs cannot be taken because it could affect the stability of the CR’s values. For this reason, figure 4 was edited to improve its contrast.
- In conclusion, the spot-welding process as a reason for the linear frequency behavior should be more emphasized.
A: Thank you for your comments. To emphasize the spot-welding process as a possible reason for the linear frequency behavior, a few lines were incorporated in the discussion.
“Nevertheless, the spot-welding process, as the cause of the linear frequency performance, cannot be discarded. For this reason, before replacing the RC-1k-D wire, the parameters of the process, like the power application time and the mechanical strain between electrodes need to be controlled.”
